# The “Dedicated” C.B.C.T. in Dentistry

**DOI:** 10.3390/ijerph20115954

**Published:** 2023-05-25

**Authors:** Salvatore Distefano, Maria Grazia Cannarozzo, Gianrico Spagnuolo, Marco Brady Bucci, Roberto Lo Giudice

**Affiliations:** 1Centro Odontoiatrico Distefano s.r.l., 95128 Catania, Italy; 2Cenacolo Odontostomatologico Italiano-Associazione Italiana Odontoiatria Generale, 95128 Catania, Italy; 3Department of Neurosciences, Reproductive Sciences and Oral Sciences, University of Naples Federico II, 80138 Naples, Italy; 4Italian Academy of Legal and Forensic Dentistry (OL-F), 19122 La Spezia, Italy; 5Department Clinical and Experimental Medicine, Messina University, 98122 Messina, Italy

**Keywords:** cone beam computed tomography C.B.C.T., D.A.P., low dose protocol, ultra low dose protocol, guidelines, medical physics

## Abstract

This position statement represents a consensus of an expert committee composed by the Italian Academy of General Dentistry (Accademia Italiana Odontoiatria Generale COI-AIOG) and Italian Academy of Legal and Forensic Dentistry (Accademia Italiana di Odontoiatria Legale e Forense OL-F) on the appropriate use of cone beam computed tomography (C.B.C.T.) in dentistry. This paper analyzes the use of C.B.C.T. in light of the rapid evolution of volumetric technologies, with the new low- and ultra-low-dose exposure programs. These upgrades are determining an improvement in the precision and safety of this methodology; therefore, the need of a guideline revision of the use of C.B.C.T. for treatment planning is mandatory. It appears necessary to develop a new model of use, which, in compliance with the principle of justification and as low as reasonably achievable (ALARA) and as low as diagnostically acceptable (ALADA), can allow a functional “Dedicated C.B.C.T.” exam optimized for the individuality of the patient.

## 1. Introduction

The evolution of volumetric technologies in dentistry is determining the need for a revision of the objective indications, in the diagnostic and operative fields, on the use of cone beam computer tomography (C.B.C.T.) [1,2,3]. The use and support of radiological imaging are of essential importance to support the dentist in the diagnostic–therapeutic decisional process with respect to biological and environmental effects.

In the dental field, the ever-increasing demand for three-dimensional investigations determines the need for an adaptation, based on updated scientific evidence, of the recommendations for performing the C.B.C.T. examination [3].

In fact, technological progress has led to a substantial decrease in irradiation, which corresponds to a better quality of information to perform the diagnostic task [1,2,3]. This allows for a more objective evaluation and an effective, safe, and precise planning of the treatment today [4,5].

Low-dose and ultra-low-dose cone beam computed tomography (C.B.C.T.) have several advantages compared to conventional 3D radiological methods. The primary advantage is the lower radiation exposure to the patient, which is particularly important for younger patients and those who require multiple scans. Traditional C.B.C.T. scanners can expose patients to radiation doses that are 10–50 times higher than conventional 2D radiological methods. Low-dose and ultra-low-dose C.B.C.T. scanners, on the other hand, can reduce the radiation dose by up to 87% compared to traditional scanners [6], if used properly; this is achieved through various techniques, such as using lower kVp and mA settings, shorter scan times, and advanced image reconstruction algorithms [2,3]. Despite the lower radiation dose, low-dose and ultra-low-dose C.B.C.T. scanners can still produce high-quality images with a similar resolution and contrast as traditional scanners. They use advanced hardware and software components, such as high-resolution detectors, adaptive filters, and iterative reconstruction algorithms, which, if well designed, can obtain better image quality than the abbreviated algorithms that can optimize the signal-to-noise ratio and reduce image artifacts [7] [see Appendix A].

Another advantage of low-dose and ultra-low-dose C.B.C.T. is that they provide high-quality, accurate 3D images of the oral and maxillofacial region. This is particularly useful for orthodontic, implant and maxillofacial surgery planning [1,5] [see Appendix A], in periodontology for the study and planning of furcation defects; in clinical and surgical endodontics; and in the evaluation of the temporomandibular joint ATM [8,9]. Low-dose and ultra-low-dose C.B.C.T. scanners are also faster and more efficient than traditional 3D radiological methods, reducing the time required for scanning and analysis. Recent evidence shows that low-dose and ultra-low-dose acquisition protocols are comparable in image quality to traditional C.B.C.T. protocols, and in the management of the diagnostic task, with a reduction in the dose equal to five times and more than the protocol C.B.C.T. recommended by the producers for the same purposes; this must lead to a rapid reflection and adaptation on the mode and recommendations for using these technologies and protocols [7,9].

The advantages of low and ultra-low-dose protocols in the use of C.B.C.T. have been evidenced in the literature, where the effectiveness of the volumetric examination for the vertical evaluation and the thickness of the bone adjacent to the anterior teeth has been evaluated as an important anatomical parameter to plan and/or monitor the effects of orthodontic or periodontal treatment. The doses and image quality were compared ex vivo with the diagnostic task of three different acquisition protocols and parameters, (low- and high-resolution) two low-dose C.B.C.T. and one normal-dose C.B.C.T. The absorbed dose was calculated using the product dose area (DAP). The Ultra low-dose C.B.C.T. protocol and the Low Dose C.B.C.T. protocol used, ejected a dose (dap 69 and 87 mGycm2) which is lower than that of panoramic views (Dap 88 mGycm2) [10]. 

Thus, with a lower absorbed dose, a higher quantity and quality of diagnostic information was obtained than those using traditional two-dimensional radiodiagnostic examinations. [See Appendix A]. The C.B.C.T. systems with low-dose protocols and ultra-low-dose C.B.C.T. are able to determine, with an appropriate diagnostic quality, both the vertical component and the horizontal component of bone, providing and representing an important accurate and reliable method for assessing the bone structure of the anterior area [10].

Low-dose and ultra-low-dose C.B.C.T. scanners can capture images in a matter of seconds or minutes, compared to several minutes or hours required by traditional scanners [3,9]. 

These machines use wider cone angles, faster rotating gantries, and optimized image acquisition protocols that can minimize patient motion and improve image registration. Additionally, the images produced by low-dose and ultra-low-dose C.B.C.T. can be easily shared and viewed by multiple practitioners, enabling more efficient communication and collaboration among specialists. Low-dose and ultra-low-dose C.B.C.T. images can be easily processed and analyzed using various software tools, such as virtual implant planning, airway analysis, and bone density measurement, that can help clinicians make more accurate and informed treatment decisions. Overall, the use of low-dose and ultra-low-dose C.B.C.T. provides many technical advantages over traditional 3D radiological methods, making it a preferred option for many dental and maxillofacial imaging needs [2,9].

Moreover, the use of digital technology has made the acquisition, storage, and transfer of C.B.C.T. images much easier and more efficient. Digital images can be easily retrieved, manipulated, and shared between dental professionals, allowing for improved collaboration and more informed treatment decisions. Furthermore, the use of digital technology has significantly reduced the need for the physical storage of radiographic images, freeing up valuable office space and reducing environmental impact.

Although the use of C.B.C.T. remains a diagnostic study that can only be used on the basis of specific criteria, its use appears justified in all those cases in which the limitations of the two-dimensional examination become clear [4,10,11,12,13,14].

In endodontics, for example, the bidimensional intraoral examination is the main radiological support in the diagnosis, planning, and execution of treatments and in the subsequent evaluation of the results [12]. However, the literature reports some limitations, and therefore, a reduction in the related diagnostic power [13], for example, in the correct and complete evaluation of the residual structure of the dental element and the surrounding bone, following pathological alterations, such as resorptions, and/or iatrogenic alterations, such as perforations [14]. 

The precise evaluation, before treatment, of the dental/alveolar complex to be recovered and of the surrounding tissues, is essential to define both the prognosis and the most suitable treatment strategy. The physical limitations of two-dimensional radiodiagnostic examination are, essentially, as follows: [1,13,14]. 

Geometric alterations, which cause disparallelisms, and consequently, image distortions;Anatomical overlays that lead to the failure to identify the actual number of root canals and their morphological configuration;Attenuation of the beam induced by the anatomical thickness of the tissues crossed.

It is essential to be aware of these limits as a radioprotection measure to prevent un-necessary repetitions of 2D intraoral examinations and to be twice as precise in identifying both the lesions and in defining the outcomes of root canal therapy, compared to the traditional intraoral examination [13]. It is possible to obtain endodontically and diagnostically valid information with appropriately managed low-dose C.B.C.T. protocols and modalities [15].

Moreover, the diagnostic limitations of two-dimensional radiographic examination are particularly relevant in certain categories of patients with cardiovascular and systemic diseases. Cardiac patients with structural problems of valves, device carriers, and/or heart prostheses are more exposed to the risk of developing infectious endocarditis (E.I.) [16]. Intervening safely and precisely is crucial when identifying possible lesions of dental elements or uncertain endodontic conditions in these patients to prevent the risk factors associated with the development of an E.I.

Nowadays, these patients can benefit, thanks to the C.B.C.T., from better diagnostic and treatment effectiveness before surgery or interventional cardiology. In addition, in these patients, the execution of a dedicated C.B.C.T. is indicated for diagnostic purposes and at the beginning of the follow-up after the cardio/valvular intervention [16].

In 2021, the American Association of Endodontists (AAE) and the American Academy of Oral and Maxillofacial Radiology (AAOMR) reached the joint conclusion that C.B.C.T. must be the reference exam in root canal retreatment and in the initial endodontic treatment of complex cases. [12,14,17] Furthermore, the additional information offered today by the C.B.C.T. examination can lead, in complex endodontic cases already defined with the two-dimensional image, to a substantial modification of the decision-making process and treatment strategy [14,18,19]. 

Patel et al. state that the 3D survey is indicated in the case of the following:Diagnosis of periapical disease in the presence of inconsistent clinical signs or symptoms;Need for diagnostic confirmation in non-odontogenic pathologies;Presence of extremely complex root canal anatomy prior to orthograde treatment;Evaluation of endodontic failures before retrograde treatment;Diagnosis of external and internal resorption;Evaluation of oromaxillofacial traumas and their treatment

A further indication for the use of C.B.C.T. is represented by the follow-up in the cases of root canal retreatments in order to objectively evaluate the effectiveness of root canal treatment and proceed, if necessary, with a timely re-operation [14,18]. Low-dose C.B.C.T. protocols have also been shown to be effective in detecting vertical fractures and perforation [19,20].

The therapeutic advantages are not limited to a single branch of dentistry, but extended to other specialties such as surgery, prosthetics, and the planning of prosthetic and implant-prosthetic treatment [5]; orthodontics as orthodontic therapy planning; oromaxillofacial combined surgery and mini screws as anchoring systems; and periodontology in complex bone and furcation defects [8]. In these disciplines, the use of 3D radiology allows for a concrete improvement in the diagnosis, planning, monitoring [20] and execution of therapy [5,14,21,22].

Furthermore, it must be considered that one of the fields of the greatest therapeutic evolution in dentistry is represented by the introduction, in clinical practice, of both static and dynamic guided therapies and patient virtualization procedures. During the diagnostic phase, these systems enable efficient, minimally invasive, rapid, precise, and effective programming, even in complex cases [23,24,25].

The result is obtained thanks to the convergence of data, derived from 3D investigations of the face (facial scanner), of the oral cavity and teeth (intraoral scan), and of the skeletal bases (C.B.C.T.) [24,25].

A further aspect, which is essential for defining the problem of using 3D radiological methods in their entirety, is represented by the need for knowledge that allows for the right balance between biological and environmental costs and the benefit of the patient [26].

Human decision-making ability and related expertise are necessary for the appropriate use of the available technology, as they are the only means of identifying the appropriate areas of use and justifications from a radiodiagnostic perspective in each individual clinical case [26,27,28].

Low-dose and ultra-low-dose protocols have been included by various manufacturers within the settings function of C.B.C.T. to assist professionals in managing optimization. However, the search for the optimal balance between image quality and dose depends on the specific clinical requirements demanded by the diagnostic task that the professional must identify and perform each time. Pre-established C.B.C.T. protocols do not always meet this requirement, which is why it is essential to acquire adequate and specialized training in relation to the use of C.B.C.T. in dental practice, as indicated in the “European Academic of Dentomaxillofacial Radiology” (EADMFR) document [17,28].

The primary objective is to produce useful clinical information with the lowest possible dose by intervening on the physical factors of the exposure load and the qualitative balance of the image.

## 2. Settings and Parameters of C.B.C.T.

There are large differences between the various C.B.C.T.s regarding the technologies and parameters for managing the dose load and image control [5]. Therefore, in addition to the protocols suggested by the manufacturer, the choice of equipment is important to allow a wide and adequate possibility of customization and optimization according to the diagnostic task [8,15,19]. There are certain parameters for managing the exposure load control and cone beam CT image quality. In detail, it is now possible to intervene on the following parameters:Pipe current (mAs). This is the product of current intensity (mA) and exposure time (s). This value adjusts the sharpness and the lightness/darkness of the image. An increase in these values increases the exposure index and the dose [2,27].Potential difference (kV) at the pipe outlet. This parameter quantifies the tissue penetration capacity and intervenes on the contrast. High kV values give rise to a more penetrating X-ray beam, with low contrast images and more details (long scale), while low values produce high contrast images (short scale).Voxel (µm). This is the constituent element of the three-dimensional physical image. Characteristic of C.B.C.T. technology is the isotropicity of the voxel. The size of the voxel can vary depending on the type of imaging exam, but typically ranges from 0.05 to 0.4 mm in size. The smaller the voxel size, the higher the resolution of the resulting 3D image, allowing for a more detailed representation of the patient’s anatomy. The size affects the spatial resolution of the image, meaning that the smaller the image, the higher the resolution. However, a decrease in the size also leads to an increase in image noise [2,27]

It is useful to know as a further element of optimization that when smaller sized voxels are used, to increase the spatial resolution, an increase in the image noise is generated; to reduce this effect, many C.B.C.T. manufacturers associate, for example, the increase in the mAs, which determines an improvement of the sharpness, with an increase in the dose load. This aspect generates the false belief that the reduction in the voxel, in an isolated manner, determines an increase in the irradiation, but in reality, it is not the size of the voxel, but the increase in the current intensity that determines the increase in the radiation doses [27].

4.Field of View (F.O.V. cm)**.** This is a term used in radiology to describe the portion of the patient’s anatomy that is captured in an imaging exam. The field of view (F.O.V.) is adjustable to capture a particular region of interest (ROI) within the volume acquisition area. The optimal F.O.V. size depends on the specific imaging exam and the clinical question being addressed [1,2,28]. The examination area should be as central as possible to the F.O.V. The width of the F.O.V. influences the effective dose emitted to the patient and should be limited to the operative diagnostic region of interest [28].

The size of the F.O.V. does not directly affect the spatial resolution of the C.B.C.T. images unless related to other management parameters, including voxel and position [29]. Since the ocular lens is a particularly radiosensitive region, it has been demonstrated that reducing the vertical extent of the field of view in order to distance it from the anatomical zone will help minimize the effective dosage [30].

As a result, to limit and minimize the radiation, it is crucial to consider the position of the F.O.V. in addition to the width.

Moreover, using narrow fields of vision with an appropriate voxel, even with low-dose protocols, is linked to higher spatial resolution in addition to dose containment, particularly when the priority is devoted to positioning the examination item as centrally as feasible within the field of view [29].

The aspect related to the centralization of the tooth inside the F.O.V. is an essential element of further optimization related to the image quality, because it manages to improve the definition of diagnostic information, such as the possibility of identifying vertical fractures (VRF), even in the presence of channel filling [18,20,29].

On the other hand, the positioning at the periphery of the F.O.V. causes the loss of the homogeneous visual definition, increasing the presence of artifacts. It is important to underline how the accuracy of the positioning of the object at the center of the F.O.V. determines a greater sensitivity and diagnostic definition than the use of the algorithms of the reduction in metal artifacts (MAR) [29,31].

Proper training and recent C.B.C.T. devices with previsualization systems, for the correct placement of the F.O.V., help with this task [32].

Furthermore, the optimal balance between the dose and the information required for diagnosis and therapy is influenced by specific technical parameters of the equipment used, which can be adjusted for some devices, such as the following:(A)The angle and rotation speed of the sensor assembly generator and exposure mode. Some equipment allows for partial or full rotations to be selected for specific acquisitions, allowing the modulation of both the patient dose and image quality. Full rotations result in a higher dose and a greater image definition, while semi-rotations reduce the dose but also the diagnostic quality. The scan speed can also affect the exposure time; higher speeds tend to reduce the dose. Additionally, most commercially available C.B.C.T.s emit pulsed exposure rather than continuous exposure, which allows for optimization by reducing the radiation [2,15,27].(B)(Filters); additional filtration at the tube outlet eliminates low-energy radiation that is not useful for diagnostic purposes, helping to further reduce the dose to the patient [2].(C)(Flat panel); the detector is the heart of the C.B.C.T. Today, all equipment has digital detectors, with either indirect or direct conversion; various types are available on the market with varying characteristics and costs. The choice of the detector influences the image quality, and therefore, all parameters related to it [2].

The new optimization possibilities linked to the evolution of radiological and information technology (IT) make it necessary to analyze and reassess the overall indications for the use of 3D C.B.C.T. in dentistry.

The innovation speed, the frequent software update that further refines and improves the function and methodologies, and artificial intelligence (A.I.) can quickly make the scientific results reported in the literature obsolete, which must be continuously updated to provide solid support for the guidelines.

Several studies show that part of the indications on the use of C.B.C.T. in certain areas of dentistry are based on opinions rather than on evidence or scientific evidence. The literature shows a lack and incompleteness of research papers on the parameter settings and clinical protocols [9,22,27].

The evaluation of more dedicated parameters, and therefore, a more appropriate use of C.B.C.T., appear to be evident when analyzing the literature and the C.B.C.T. use in different fields of dentistry.

In orthodontics, the technological evolution and recent evidence show that the radiation of an ultra-low-dose C.B.C.T. (ULD-UL) with lateral cephalometric reconstruction (RLC) can deliver an effective dose with a reduction of 87% compared to standard C.B.C.T. protocols, resulting in an examination with 11–18 microSieverts (μSv).

A combination of traditional 2D examinations, orthopantomography (PAN), and lateral cephalometric (SLC) results in a dose of 27–30 (μSv), which is therefore higher than the 11–18 (μSv) dose of the ULD-UL C.B.C.T. [6].

In terms of radioprotection, it is possible to reduce the radiant impact suffered by the patient, improving the visibility and analysis of anatomical structures for diagnostic and treatment purposes, such as the management of impacted canines [6].

This evaluation is in contrast to what is reported, for example, in the Italian text, “Recommendations for the correct use of volumetric TC equipment “Cone Beam”, regarding the amount of the effective dose expressed for C.B.C.T., PAN, and LC examinations currently in force [33]. 

Contact proximity with nearby anatomical structures (roots of contiguous elements) can lead to root resorption. In certain circumstances, the European guidelines recommend the integration of 2D images with C.B.C.T. exams optimized with a reduced F.O.V. in order to allow an adequate and complete view of the anterior maxillary region and evaluate the impacted teeth with the surrounding structures, allowing an accurate diagnosis and surgical/orthodontic treatment [1,6,10].

It should be noted that missed diagnosis or late treatment can lead to root resorption of the permanent adjacent incisors in 48% of cases, often complicating the outcome of orthodontic treatment and the prognosis of the dental elements interested [6].

In implant dentistry, C.B.C.T. is recommended as a diagnostic tool for presurgical planning and oral implant-related diagnosis. Compared to 2D imaging, C.B.C.T. is more accurate in predicting implant dimensions and the need for bone grafting procedures. Low-dose C.B.C.T. (LD-C.B.C.T.) protocols produce acceptable 3D images while substantially reducing the patient’s radiation exposure [24,26]. The effective dose can be adjusted by selecting appropriate scanning parameters and fields of view (F.O.V.). In guided surgery, satisfactory results were obtained by adapting the milliamperage setting and exposure time to achieve low-dose C.B.C.T. protocols [26,27]. Additionally, radiation exposure can be reduced by minimizing the F.O.V. (4 × 4–8 × 8 cm^2^) and potentially combining minimal fields. As a result, effective doses can range widely for C.B.C.T., from 5 μSv to 1073 μSv [1,5,27].

In oral surgery, in the surgical treatment of impacted teeth such as canines and in complex extraction procedures of third molars, the use and prescription of the C.B.C.T. is a commonly encountered event, representing a frequent indication; numerous evidences demonstrate how it is important and useful to significantly limit the irradiation to the patient, and to use adequate exposure parameters of F.O.V. as small as 4 × 4 or smaller [1,17,32]. This has proven to be a favorable optimization strategy in surgical management. It is advisable to reserve the largest fields of view only in the presence of associated pathology, which involves the tissues adjacent to the tooth, such as, for example, the presence of voluminous lesions [5]. The low-dose C.B.C.T., especially if associated with a F.O.V. restricted with voxels not below 0.2 mm, can be considered a valid solution in the planning of surgical procedures [1,10,32].

Some of the limitations linked to patient and environmental radioprotection criteria, applied as different standards for different states that regulate the use of C.B.C.T., are now based on outdated recommendations.

Further research that takes into account the better management of exposure parameters and image quality linked to technological evolution is necessary to obtain objective data on the clinical use of 3D diagnostics technologies [27,28].

The primary objective is to produce useful clinical information with the lowest possible dose, by intervening on the physical factors of the exposure dose and the qualitative balance of the image [2]. This requires a careful evaluation of the clinical indication and a thorough understanding of the technology and its limitations [5,28,34]. The decision to use C.B.C.T. should always be guided by the principle of doing no harm and ensuring that the benefits to the patient outweigh any potential risks or costs.

The evolution of volumetric technologies in dentistry should lead to a revision of the objective indications for the use of C.B.C.T.

The technology offers several advantages over traditional 2D imaging, including improved visualization of complex anatomy, improved diagnostic accuracy, and the ability to visualize 3D structures. However, the use of C.B.C.T. should always be guided by a careful assessment of the clinical requirements and a thorough understanding of the technology and its limitations [14,27,28].

This means that the imaging protocols should be tailored to the specific needs of each patient, taking into account factors such as their size, anatomy, and the type of treatment they are undergoing. This will result in a more efficient and effective exam that is optimized for the patient’s individual needs [28]. Adequate and specialized training in the use of C.B.C.T. is essential to ensure that the technology is used appropriately and effectively following the principles of justification and as low as reasonably achievable (ALARA) and as low as diagnostically acceptable (ALADA) [35].

ALARA (as low as reasonably achievable) is a principle in medical imaging that aims to minimize a patient’s exposure to ionizing radiation by using the lowest effective radiation dose necessary to produce an accurate and diagnostic image. The goal of ALARA is to reduce the risk of adverse effects from radiation exposure while still obtaining the necessary diagnostic information. The concept of ALARA is a central principle of a general nature in radioprotection; it promotes and is essentially based on the importance of optimization processes and is based on the hypothesis of a linear dose-effect relationship without threshold (evaluates the stochastic) [36]. It already contains all the potential to exhaustively express the radiation protection concept applied to medical diagnostic imaging as per national and international directives [36]. It is a principle that has been used for more than 20 years in the rare radiology field in order to obtain valid diagnostic information using the lowest reasonably possible dose (it is also used more extensively in other professional fields for the prevention of radiation in exposed populations) [36,37].

ALADA (as low as diagnostically acceptable) is a similar concept in medical imaging, but focuses on ensuring that the radiation dose used is low enough to minimize risks, but high enough to produce a diagnostic image. ALADA aims to express its optimization contribution exclusively to the medical/dental activity, linked to diagnostic acceptability. In other words, ALADA balances the need for diagnostic quality with the need for radiation safety [37].

Both ALARA and ALADA are important principles in radiology, and radiologists strive to achieve these goals in their practice by using appropriate imaging techniques, optimizing equipment settings, and taking other steps to minimize the patient’s exposure to ionizing radiation.

For the purposes of correct radiological practice, it is also necessary to highlight how it is propaedeutic to preliminarily justify any radiological exposure for a specific diagnostic/therapeutic purpose, taking into account the individual characteristics, the patient, and the techniques which involve a minor exposure to ionizing radiation, as defined by the European directive 59/2013 art. 55 [37].

## 3. Regulatory and Legal Aspects Related to the Use of C.B.C.T. in Dentistry

It becomes necessary to ensure the patient the best diagnostic and therapeutic quality but also the best safety obtainable at the present moment. In Italy, the law in force on healthcare professional liability puts the safety of care and of the assisted person in first place [38].

With reference to the European community regulations on radiation protection (which Italy has implemented in Legislative Decree 101/2020), any patient exposure to ionizing radiation for medical purposes must be subject to the principle of justification (assessment of potential diagnostic or therapeutic advantages—art. 157) and the principle of optimization (exposing the patient to the lowest level of radiation reasonably obtainable and compatible with the achievement of the required diagnostic information—art. 158) [37,38].

Furthermore, all new types of practices involving medical exposures must be preliminarily justified before being generally adopted and must be reviewed whenever new and relevant evidence about their efficacy is acquired [37,38].

It is therefore obvious that the justification process must be considered in continuous evolution, above all, in consideration of the diffusion of new technologies which involve an evolution of the modalities of access to diagnostic information, as in the case of the 3D C.B.C.T. The Italian ministerial “Recommendations for the correct use of Cone Beam volumetric CT equipment” date back to May 2010 [33], and the data contained in this article make clear the need for a regulatory review with the main objective of adapting and updating the correct use of available scientific knowledge for improved patient safety.

## 4. Conclusions

It appears to be evident that there is a necessity to update the recommendations for the C.B.C.T use and to improve knowledge and training to fully exploit the potential and the benefits that the technological evolution has given to the clinician [3,27]. Consequently, it appears necessary to develop a new model of use, which, in compliance with the principle of justification and as low as reasonably achievable (ALARA) and as low as diagnostically acceptable (ALADA), can allow a functional “Dedicated C.B.C.T.” exam that is optimized for the individuality of the patient, for treatment planning, and for the improvement of precision and safety of the therapeutic target [3,4,9,18,19,22,25,32,39,40].

## Data Availability

Original data are available upon request to the corresponding author.

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
