# Peer review of "The “Dedicated” C.B.C.T. in Dentistry"

_ijerph, 2023, doi:10.3390/ijerph20115954_

Round 1

Reviewer 1 Report

 Would you please detail more the advantages of  using low-dose/ ultra low-dose CBCT compared to current conventionals 3D radiological methods.

Author Response

A paragraph have been added in text to detail the advantages of  using low-dose/ ultra low-dose CBCT

Reviewer 2 Report

The title of the manuscript is ‘The “dedicated” CBCT in dentistry.’ I feel the term “dedicated” is misguiding and not conveying the true meaning. The authors can choose word such as “customized”.

In the manuscript, authors justify the need for customization of the exposure parameters while performing the CBCT scan to minimize the patient’s exposure to ionizing radiation based on conditions such as the size of the patient, anatomy of the area to be scanned, and type of treatment the patient is undergoing.

Authors fail to clearly highlight how “dedicated” CBCT will reduce the patient’s exposure to ionizing radiation over and above the existing as low as reasonably achievable (ALARA) and as low as diagnostically acceptable (ALADA) guidelines.

The manuscript is riddled with statements that are either grammatically incorrect or incomplete and could have been written in a way that is easy to understand from a general reader’s perspective.

In Abstract Line 27, short forms are used: Alara/Alada

Line 53, 60, 66, 103, 112, 142, 176 need to be rewritten.

Line 74, AAOMR is outside brackets.

Line 80, Patel and Al is incorrect, it should be Patel el al.

Line 122, the cited reference is incorrect.

Line 151, 160, 165 are showing formatting errors.

Line 172, the universally accepted short form of Artificial Intelligence is AI.

Line 185-186, give reference to the said European guidelines

Line 187, FOV should be used instead of fov.

The paragraph (Line 183-192) is out of place. Giving an elaborate example of an impacted canine is not in line with the flow of the manuscript.

Line 204-207, give proper citation.

 Line 221, The evolution of volumetric technologies in dentistry should lead to a revision of the objective indications for the use of CBCT.

 Even though this manuscript represents a consensus of an expert committee consisting Italian Academy of Forensic Legal Dentistry as its part, forensic implications of the “dedicated” CBCT scan are not addressed.

The addition to the existing knowledge is negligible. Only lines 228-231 address the title and objective of the manuscript.  

This work is not a significant contribution to the already existing and universally followed principle of ALARA/ALADA. 

Author Response

Dear reviewer many thanks for your suggestion that helped to deeply improve the manuscript. 

following a detailed point to point answer to your suggestion

The title of the manuscript is ‘The “dedicated” CBCT in dentistry.’ I feel the term “dedicated” is misguiding and not conveying the true meaning. The authors can choose word such as “customized”.

Personalized usually refers to the person; with the term "Dedicated" we refer, in addition to the patient, to an extensively specific use of the cb3d in the dental field.

Regarding the suggestion of the name ''customized'' A good justification of the CBCT practice already implies a "customization" that is, it must take into account both the specific objectives of the exposure and the individual characteristics of the person, according to what is foreseen in the existing legal system in Europe, the European Directive 59/2013. art 55.

A paragraph have been added in text

2)

Authors fail to clearly highlight how “dedicated” CBCT will reduce the patient’s exposure to ionizing radiation over and above the existing as low as reasonably achievable (ALARA) and as low as diagnostically acceptable (ALADA) guidelines.

1) We identify in our contribution seven technically detailed points, on how to act in relation to the management parameters in order to reduce the dose and balance the image quality according to a dedicated use of the CBCT.

2) We have also implemented, further, in the contribution, the role of the FOV in the optimization process, defining the importance of its use beyond the breadth in clinical use

3) The existing guidelines do not indicate the management parameters: we propose a new vision of the problem, exposing in their entirety all the parameters on which it is possible to intervene to reduce doses and obtain valid images.

4) Alara and Alada are not guidelines, but respectively principles and concepts of radiation protection (we will specify these aspects in answer 5) which concern the optimization of diagnostic imaging procedures, to which the guidelines implicitly refer.

3)

Even though this manuscript represents a consensus of an expert committee consisting Italian Academy of Forensic Legal Dentistry as its part, forensic implications of the “dedicated” CBCT scan are not addressed.

A paragraph have been added in text

4)

The addition to the existing knowledge is negligible. Only lines 228-231 address the title and objective of the manuscript.

As many scientific evidence underline, the competence, training and evidence in the literature regarding the use of CBCT in dentistry are lacking, our work in its entirety aims to help improve knowledge regarding the new possibilities of managing in-depth analysis radiodiagnostic cbct.

This work is not a significant contribution to the already existing and universally followed principle of ALARA/ALADA. 

Alara and Alada are respectively radioprotection concepts and principles that pertain exclusively to the optimization of the examination, our dedicated cbct contribution pertains both to the principle of justification of the examination (improve the precision of the treatment and of the therapeutic objective) and to optimization (analyze and stimulate in-depth study of the technical parameters that govern the dose/image quality balance.)               

The concept of ALARA is a central principle of a general nature in radioprotection, it promotes and is essentially based on the importance of optimization processes, it is based on the hypothesis of a linear dose-effect relationship without a threshold (evaluates the stachastic risk); it already contains all the potential to exhaustively express the radiation protection concept applied to medical diagnostic imaging as per national and international directives. It is a principle used for more than 20 years in the rare radiological field in order to obtain valid diagnostic information with the lowest reasonably possible dose, (it is also used more extensively in other professional fields for the prevention of radiation in exposed populations). The reference concerning the optimization of European directive 59/2013 art.56

Alara's optimization principle does not express technical details related to the use of cbct and does not concern the justification of the exam.

Alada aims to express its optimization contribution exclusively to the medical/dental activity linked to diagnostic acceptability.

Our contribution (dedicated cbct) wants to express in harmony with the above radiological protection concepts, a model of use based on the knowledge of the correct applications (justification) of the cb3d exam and on the knowledge of the main technical parameters of exposure load management and control quality of the image (optimization) which must take into account the specific needs required by the precise diagnostic/operative task, underlining the importance of the correct understanding of the possibilities that technological evolution offers us in the field of dental volumetric radiodiagnostics. Especially for the possibilities today of using radiological devices that allow us to obtain CBCT images and volumetric reconstructions adequate for therapeutic purposes and purposes, with a considerably lower irradiation than the standard CBCT protocols, taken into consideration by the current recommendations for use. The information collected in this way can be oriented not only to diagnostic purposes but can be used specifically as a direct aid to treatment planning and assistance, being able to improve the definition of the operative path by determining targeted interventional actions, improving the precision and safety of the therapeutic objective in the interest and protection of the patient's health.

Therefore in line with both the principle of justification of the examination and with the principle of optimization, focusing on current and future developments and on the benefits that the correct use of this technology can offer.

We've added content about that.

In Abstract Line 27, short forms are used: Alara/Alada

The text have been corrected

Line 53, 60, 66, 103, 112, 142, 176 need to be rewritten.

The lines have been rewritten

Line 74, AAOMR is outside brackets.

Corrected

Line 80, Patel and Al is incorrect, it should be Patel el al.

corrected

Line 122, the cited reference is incorrect.

Corrected

Line 151, 160, 165 are showing formatting errors.

Line 172, the universally accepted short form of Artificial Intelligence is AI.

 Corrected

Line 185-186, give reference to the said European guidelines

Referenced added

Line 187, FOV should be used instead of fov.

Corrected

The paragraph (Line 183-192) is out of place. Giving an elaborate example of an impacted canine is not in line with the flow of the manuscript.

Line 204-207, give proper citation.

Citation added

 Line 221, The evolution of volumetric technologies in dentistry should lead to a revision of the objective indications for the use of CBCT.

Text corrected

Reviewer 3 Report

The authors presented an interesting article related to the use of CBCT in dentistry.

It is however limited in the cited applications to endodontics and orthodontics, while one of the most important indication for CBCT is implant dentistry... I would therefore recommend further elaborating on that.

Author Response

A paragraph have been added to underline the importance of cbct and low dose protocol in implant dentistry

Moreover the whole section describing the importance in many dentistry field have been improved

Round 2

Reviewer 3 Report

Thank you for the corrections. I have no further comments.